# An Analysis of Loss Functions for Heavily Imbalanced Lesion Segmentation

**DOI:** 10.3390/s24061981

**Published:** 2024-03-20

**Authors:** Mariano Cabezas, Yago Diez

**Affiliations:** 1Brain and Mind Centre, The University of Sydney, Camperdown, NSW 2050, Australia; 2Faculty of Science, Yamagata University, Yamagata 990-8560, Japan; yago@sci.kj.yamagata-u.ac.jp

**Keywords:** medical imaging sensors, magnetic resonance imaging, brain, lesion segmentation, imbalanced dataset, loss functions

## Abstract

Heavily imbalanced datasets are common in lesion segmentation. Specifically, the lesions usually comprise less than 5% of the whole image volume when dealing with brain MRI. A common solution when training with a limited dataset is the use of specific loss functions that rebalance the effect of background and foreground voxels. These approaches are usually evaluated running a single cross-validation split without taking into account other possible random aspects that might affect the true improvement of the final metric (i.e., random weight initialisation or random shuffling). Furthermore, the evolution of the effect of the loss on the heavily imbalanced class is usually not analysed during the training phase. In this work, we present an analysis of different common loss metrics during training on public datasets dealing with brain lesion segmentation in heavy imbalanced datasets. In order to limit the effect of hyperparameter tuning and architecture, we chose a 3D Unet architecture due to its ability to provide good performance on different segmentation applications. We evaluated this framework on two public datasets and we observed that weighted losses have a similar performance on average, even though heavily weighting the gradient of the foreground class gives better performance in terms of true positive segmentation.

## 1. Introduction

Magnetic resonance imaging (MRI) is one of the most common techniques using sensors to obtain imaging information from the brain. Furthermore, lesion segmentation is the most common image analysis task applied to brain MRI for different neurological pathologies, such as multiple sclerosis [1,2], Alzheimer’s disease [3] or vascular cognitive impairment [4]. Brain lesions, usually identified as white matter hyperintensities on T2-weighted images, represent a small percentage of the whole brain volume, can have different shapes and volumes and their spatial distribution can greatly vary between patients. Due to these issues, the accurate detection and segmentation of white matter hyperintensities is a challenging segmentation task.

Recently, deep learning approaches have rapidly become the state of the art for medical image analysis [5,6], including binary image segmentation [7]. However, a severe data imbalance can bias the training of the models and produce unexpected results (i.e., reduce the accuracy of the lesion segmentation) [8,9]. The most common solutions are the use of sampling techniques to increase the number of positive samples [10] and loss functions that give different weights to the foreground and background classes, either implicitly [11,12] or explicitly [13]. While different losses have been presented for medical imaging [14], they are rarely analysed for datasets with heavy imbalances. Specifically, their effect on the underrepresented class is not properly understood, how the loss evolves through time per epoch is rarely studied, and the effect of randomness is usually excluded from the results, even though weight optimisation is inherently a stochastic process. To that end, multiple studies have been published on understanding prediction uncertainty through probabilistic frameworks and deep ensembles [15,16,17,18], suggesting the importance of reporting results with multiple runs and random seeds. Furthermore, losses are usually defined based on their value, but they are rarely designed taking into account their derivative and their effect on optimisation.

In this paper, we present an analysis of the most common losses used for imbalanced datasets, and we study their evolution during training focusing on the effect on the underrepresented class. Furthermore, we propose a new weighted loss function based on the gradient of the cross-entropy loss (as opposed to focusing only on the loss function alone). The rest of the paper is structured as follows: in Section 2, we present the framework and public datasets we used, followed by the results in Section 3, a short discussion in Section 4 and conclusions and future works in Section 5.

## 2. Materials and Methods

### 2.1. Public Datasets

For this analysis, we used two public datasets with manual annotations of brain lesions and a heavy imbalance. The statistics of the number of foreground and background voxels are summarised in Table 1 and examples of images are given in Figure 1. While the ratio between foreground and background voxels is bigger inside the training patches, they still represent less than 5% of the whole patch on average.

#### 2.1.1. White Matter Hyperintensities (WMH) Challenge 2017 [4]

The WMH Segmentation Challenge was held during the MICCAI 2017 conference, and it provided a standardized assessment of automatic methods for the segmentation of WMH. The task for the challenge was defined as the segmentation of WMH of presumed vascular origin on brain MR images. A total of 60 labelled images were acquired from three different scanners with different sensors from three different vendors, in three different institutes. For each subject, a 3D T1-weighted and a 2D multi-slice FLAIR image were provided. Details on the acquisition and preprocessing steps for these images are summarised in the original challenge paper [4]. In brief, all images were manually labelled for white matter hyperintensities by an expert, and these annotations were afterwards reviewed by a second expert who created the final mask. Other pathological regions were also labelled. However, these regions were not evaluated during the challenge and, thus, we ignored this label for our analysis. Finally, T1-weighted images were co-registered to the FLAIR space during pre-processing, and we used these pre-processed images for our experiments.

The parameters of each acquisition are summarised as follows:UMC Utrecht, 3 T Philips Achieva: 3D T1-weighted sequence (192 slices, slice size: 256 × 256, voxel size: 1.00 × 1.00 × 1.00 mm^3^, repetition time (TR)/echo time (TE): 7.9/4.5 ms), 2D FLAIR sequence (48 slices, slice size: 240 × 240, voxel size: 0.96 × 0.95 × 3.00 mm^3^, TR/TE/inversion time (TI): 11,000/125/2800 ms).NUHS Singapore, 3 T Siemens TrioTim: 3D T1-weighted sequence (192 slices, slice size: 256 × 256, voxel size: 1.00 × 1.00 × 1.00 mm^3^, TR/TE/TI: 2, 300/1.9/900 ms), 2D FLAIR sequence (48 slices, slice size: 256 × 256, voxel size: 1.00 × 1.00 × 3.00 mm^3^, TR/TE/TI: 9000/82/2500 ms).VU Amsterdam, 3 T GE Signa HDxt: 3D T1-weighted sequence (176 slices, slice size: 256 × 256, voxel size: 0.94×0.94×1.00 ${\mathrm{mm}}^{3}$, TR/TE: 7.8/3.0 ms), 3D FLAIR sequence (132 slices, slice size: 83 × 256, voxel size: 0.98×0.98×1.20 ${\mathrm{mm}}^{3}$, TR/TE/TI: 8000/126/2340 ms).

#### 2.1.2. Ljubljana Longitudinal Multiple Sclerosis Lesion Dataset [1]

This database contains baseline and follow-up MR images of 20 multiple sclerosis (MS) patients. The images were acquired on a 1.5 T Philips MRI machine at the University Medical Centre Ljubljana (UMCL), and the data were anonymized [1]. Each patient’s MR acquisition contained a 2D T1-weighted (spin echo sequence, repetition time (TR) = 600 ms, echo time (TE) = 15 ms, flip angle (FA) = 90°, sampling of 0.9×0.9×3 mm with no inter-slice gap resulting in a 256×256×45 lattice), a 2D T2-weighted (spin echo sequence, TR = 4500 ms, TE = 100 ms, FA = 90°, sampling of 0.45×0.45×3 mm with no inter-slice gap, resulting in a 512×512×45 lattice), and a 2D FLAIR image (TR = 11,000, TE = 140, TI = 2800, FA = 90, sampling of 0.9×0.9×3 mm with no inter-slice gap, resulting in a 256×256×49 lattice). The median time between the baseline and follow-up studies was 311 days, ranging from 81 to 723 days with the interquartile range (IQR) of 223 days. For our analysis, we only focused on the FLAIR images that were skull stripped and bias corrected using N4, followed by coregistration to the follow-up space using rigid registration. To create a ground truth mask, an image neuroanalyst expert at our group labelled all positive activity (new and enlarging lesions) using the coregistered FLAIR images and their subtraction. Furthermore, 3 of the subjects were excluded due to a lack of positive activity after the labelling. Therefore, only 17 patients from the original dataset were used for the experiments.

#### 2.1.3. Data Preparation

For both datasets we extracted patches of size 32×32×32 using a sliding window (with an overlap of 16 voxels) on the brain bounding box to minimise the number of voxels that did not belong to the brain. These patches were then sampled to decrease the imbalance between foreground and background voxels. Only patches containing lesion voxels were used for training in groups of 16.

### 2.2. Network Architecture

In the last few years, the 3D Unet architecture [19] has become the state of the art in volumetric medical image segmentation for different tasks and images [7]. In order to train all the losses with a robust architecture, we decided to use a simple 3D Unet of 2,906,913 parameters using basic convolutional blocks (with 32, 64, 128 and 256 kernels of size 3×3×3), followed by ReLU activation, group norm (with groups of 8 channels) and max pooling of size 2 (see Figure 2). These hyperparameters were chosen empirically based on the best performance of our previous works on other private datasets for lesion segmentation [20,21]. To guarantee a fair comparison, these hyperparameters were also shared for all the experiments (different seeds and loss functions). The weights were randomly initialised using the same seed for all the losses in a given experiment to reduce the effect of random initialisation. This seed was also used to ensure that the training batches per epoch were shuffled the same way for all the losses.

### 2.3. Loss Functions

#### 2.3.1. Cross-Entropy

The cross-entropy loss (**xent**) is one of the most commonly used loss functions for deep neural networks. If we consider the distribution of the real labels ($y_{i}$) and the predicted ones ($\hat{y_{i}}$), the cross-entropy expresses the similarity of these two distributions in the following form:(1)$$L_{CE}\left({\hat{y}}_{i},y_{i}\right)=\left\{\begin{array}{cc} -\log{\hat{y}}_{i}, & \;\mathrm{if}\;\mathrm{foreground}\\ -\log(1-{\hat{y}}_{i}), & \;\mathrm{if}\;\mathrm{background} \end{array}\right.$$

In our case, this equation represents the loss for a single voxel, and the final value is computed as the mean of the loss for all the voxels. By definition, all voxels have an equal weight; therefore, during training, the optimiser will tend to favour the majority class.

#### 2.3.2. Focal Loss

The focal loss function was presented by Tsung-Yi Lin et al. [13] for dense object detection in the presence of an imbalance between positive and negative samples. Through the use of a weighting variable (α) and a modulating factor (γ), they extended the cross-entropy loss to increase the importance of positive samples and reduce the effect of samples correctly classified with a high confidence. Following the notation for the cross-entropy loss, the focal loss for a prediction $\hat{y_{i}}$ can be defined as:(2)$$L_{FL}\left({\hat{y}}_{i},y_{i}\right)=\left\{\begin{array}{cc} -\alpha {\left(1-\hat{y_{i}}\right)}^{\gamma }\log\hat{y_{i}}, & \;\mathrm{if}\;\mathrm{foreground}\\ -\left(1-\alpha \right){\hat{y_{i}}}^{\gamma }\log\left(1-\hat{y_{i}}\right), & \;\mathrm{if}\;\mathrm{background} \end{array}\right.$$

For our experiments, we used γ=2 as suggested on the paper, and two different α values: $\alpha_{1}=0.25$ (**focal1**) and $\alpha_{2}=0.75$ (**focal2**). On the original paper, they suggest to decrease α as γ is increased, and they noted that α=0.25 gave the best results with γ=2. However, we also wanted to test the loss with a higher α value to give a higher weight to positive samples.

#### 2.3.3. Generalised Dice Loss

The generalised Dice loss was presented by Sudre et al. [12] to balance the effect of positive and negative samples, based on a generalisation of the Dice similarity metric (DSC) [22]. This loss function can be defined as:(3)$$L_{gDSC}\left(\hat{Y},Y\right)=1-2\cdot \frac{\left(w_{fg}+w_{bg}\right)\cdot \sum_{i}^{N}{\hat{y}}_{i}\cdot y_{i}+w_{bg}\cdot (N-\sum_{i}^{N}\left({\hat{y}}_{i}+y_{i}\right)}{\left(w_{fg}-w_{bg}\right)\cdot \sum_{i}^{N}\left({\hat{y}}_{i}+y_{i}\right)+2\cdot Nw_{bg}},$$
where $\hat{Y}$ and *Y* represent the predicted and true segmentation for the *N* image voxels (${\hat{y}}_{i}$ and $y_{i}$), and $w_{bg}$ and $w_{fg}$ are weights for the background and foreground class, respectively.

In the special case where $w_{bg}=0$, the generalised Dice loss becomes the commonly used binary Dice loss originally presented by Milletari et al. [11]:(4)$$L_{DSC}\left(\hat{Y},Y\right)=1-2\cdot \frac{\sum_{i}^{N}{\hat{y}}_{i}\cdot y_{i}+\epsilon }{\sum_{i}^{N}\left({\hat{y}}_{i}+y_{i}\right)+\epsilon },$$
where ϵ is a small smoothing constant to avoid numerical issues when dividing by 0. In our experiments, we used the generalised loss (**gdsc**) as defined on the original paper ($w_{bg}=1/N_{bg}^{2}$ and $w_{fg}=1/N_{fg}^{2}$, where $N_{bg}$ and $N_{fg}$ refer to the number of background and foreground voxels), the binary Dice loss (**dsc**) ($w_{bg}=0$ and $w_{fg}=1$) and a combination of the binary Dice loss and cross-entropy with equal weights (**mixed**).

#### 2.3.4. Weighted Gradient Loss

Similarly to the focal loss, we propose to introduce a weighting factor and a modular factor to the cross-entropy loss. However, instead of applying that factor on the loss function itself, we propose to apply it to the gradient of the cross-entropy when combined with the gradient of the sigmoid function as follows:(5)$$\frac{\partial L_{wCE}\left({\hat{y}}_{i},y_{i}\right)}{\partial f_{i}}=\left\{\begin{array}{cc} \alpha {\left(1-{\hat{y}}_{i}\right)}^{\gamma }, & \;\mathrm{if}\;\mathrm{foreground}\\ \left(1-\alpha \right){\hat{y}}_{i}^{\gamma }, & \;\mathrm{if}\;\mathrm{background} \end{array}\right.$$
where $f_{i}$ represents the output of the last layer before applying the final sigmoid activation. In our experiments, we set γ=2 and $\alpha =N_{bg}/N$ (**new**).

### 2.4. Experimental Design

In order to counter the effect on random initialisation and shuffling, we developed a framework to train the same network with the same initialisation and batches by setting the same random seed for each loss for a given experiment. With that setup, we run 5 different 5-fold cross-validation experiments with the same training/testing splits. We then evaluate the models at the end of each epoch for a maximum of 25 epochs by analysing the Dice similarity coefficient ($DSC=2*\frac{TP}{2*TP+FN+FP}$); sensitivity, also known as true positive fraction or $TP_{f}$ for short ($TP_{f}=\frac{TP}{TP+FN}$); and precision ($P=\frac{TP}{TP+FP}$). We compute all three metrics for the training patches, and the training and testing image segmentations reconstructed from patches. In brief, patches of 32 × 32 × 32 are uniformly sampled for each image with a sliding window of 16 × 16 × 16, and the final prediction for each voxel is averaged between all the overlapping results. For the binary metrics, we first binarise the predicted output of the network (with a threshold of 0.5). The goal of this setup is to analyse whether the difference in different executions are actually due to the training variance caused by randomisation or the losses themselves.

### 2.5. Implementation Details

The framework was implemented using Python 3.6.9 and pytorch version 1.5.0 and the code is publicly available at https://github.com/marianocabezas/rethinking_dsc (accessed on 1 January 2024). The whole analysis was run on a NVIDIA DGX-1 server with 80 CPU cores, 504 GB of RAM and two Tesla V100-SXM2.

## 3. Results

To analyse the performance of the different trained models for each epoch and seed, we used the $TP_{f}$ and precision to illustrate the balance between over- and undersegmentation and the DSC to highlight the overall segmentation. Ideally, we would like to maximise all metrics, but in practise, increasing the $TP_{f}$ also increases the false positives, affecting the precision and DSC values. Table 2 summarises the best performance when the best epoch and random seed are known, while Figure 3 illustrates the curves for all folds and seeds for each dataset in terms of each of the three different metrics we analysed (plots for each fold are detailed in Figure 4 and Figure 5). The curves are separated by the type of data being evaluated (input patches and whole image segmentations reconstructed from patch segmentations) and represent a summarised version of all the metrics for each different training seed setup. The mean metric value is represented with a solid coloured line, while error bands around that line are bounded by the minimum and maximum value per epoch for all the seeds. By using upper and lower bounds, we can represent stochastic variability that could lead to different conclusions if different seeds are used for each method. Furthermore, the error bands also illustrate how robust the models can be to the problems represented by each dataset.

Observing the table and curves, there is a clear discrepancy between the performance on the training patches and the overall segmentation for the whole image. This difference is emphasised on the Dice-based losses. Furthermore, it is also clear that weighting the foreground class, either explicitly (focal and weighted loss) or implicitly (Dice based), improves the performance in terms of the final DSC metric. However, the larger the weight, the better the results. Furthermore, a high weight (higher than 0.9 for the weighted loss) has a direct impact on the $TP_{f}$ measure throughout the whole training process.

A closer analysis of the curves highlights the trade-off between the precision and $TP_{f}$ segmentation measures (also illustrated in the qualitative examples from Figure 4 and Figure 5). As the training progresses, the $TP_{f}$ metric decreases slowly, while the precision improves (leading to an overall slightly better DSC value). Furthermore, if we observe the bands for the losses, there is a clear overlap between all of them for the DSC metric. This suggests that there is a large variability in the model performance depending on the chosen random seed and fold. In general, the observed training pattern is similar for the weighted losses (Dice based, focal and weighted) with an initial higher peak in terms of $TP_{f}$ and a slower decrease for the losses with a higher weight for the foreground class. While the behaviour is similar for both datasets, there is a bigger difference between losses for the longitudinal dataset, which also presents a higher imbalance between classes. In fact, the variability in the LIT dataset is also larger (wider error bands), leading to a larger overlap and an increased difficulty to separate between methods. Curves for the WMH dataset, on the other hand, have thinner error bands, and there is a clear separation between our proposed loss function and the rest of methods for the $TP_{f}$ and precision metrics on the first few epochs, even though the DSC values remain similar for the training and testing images. Nonetheless, the curves converge to similar results as the number of epochs trained increases. Finally, the curves on Figure 4 and Figure 5 show the same trends observed on the general dataset curves, albeit with reduced error bands (fewer samples are analysed per fold). Analysing folds carefully (each row on the figures), the LIT shows larger differences between folds. Specifically, the first row shows overall higher $TP_{f}$ and DSC values for all methods when compared to the other folds.

To complement the analysis of the curves, qualitative results for epoch 25 (the last epoch) are provided in Figure 6 and Figure 7 for the LIT and WMH dataset, respectively. While there seems to be a general agreement in terms of true positives (green colour), most losses and examples show an increase in false negatives (blue colour) when compared to the new proposal (as also evidenced by the numerical results). In particular, small lesions tend to be missed by losses that do not heavily weight positive predictions, with cross-entropy having the lowest detection. This is contrasted by a decrease in the number of false positives (red colour), also evidenced by the precision metric on the numerical results. From the Dice-based losses, mixed has the overall best results, with the dsc loss having a large number of false positives (especially on the LIT dataset). Comparing the two focal losses, the results are fairly similar with small differences in detection depending on the case, suggesting that the value of α might not be as important as expected.

Finally, we summarise the DSC, $TP_{f}$ and precision metrics on the training and testing images using a violin plot in Figure 8 to analyse the distribution of these metrics for each loss according to the random seeds. Once again, we can observe that the best performance (highest bound) is obtained with the heavily weighted gradient loss (new), even though the plots present a similar shape for DSC in both datasets for most of the losses and a large variability in general. However, we can also observe some important differences on the $TP_{f}$ distribution for both datasets. On the longitudinal dataset, the focal losses and the cross-entropy present a larger concentration of lower $TP_{f}$ values when compared to the other losses, while only the lightly weighted focal loss (focal1) and cross-entropy present a lower performance on the WMH dataset. These two losses also present the worst overall performance on the longitudinal dataset in terms of DSC (as illustrated by a longer and spread plot, concentrated on the lower bound).

## 4. Discussion

### 4.1. The Effect of Confident Errors on the Loss Function

The Dice loss is one of the most commonly used techniques to address class imbalance due to its application as a metric for segmentation [23]. Furthermore, it is usually believed that it is one of the best choices for heavily imbalanced datasets. However, the original metric was not developed for optimisation and was designed to address a set theory problems (specifically, to measure the overlap between two sets). As a consequence, there are certain unexpected side effects when pairing it with common activations for segmentation (sigmoid and softmax). One of these unexpected side effects is the effect of confidently wrong predictions (for a mathematical analysis through derivation, we refer the reader to Appendix A). This effect is evidenced by the metric curves for the Dice loss where we found the largest discrepancy between patch-based metrics (the ones during training) and image-based metrics (the final desired output). On the first step, the Dice loss favours the prediction of positive samples, leading to a high true $TP_{f}$ at the cost of a large number of false positive predictions (as evidenced by the low DSC value). Some of these erroneous predictions maintain a high confidence score throughout the whole process, leading to false positive predictions that are never corrected even with a large number of epochs. These findings align with another recent study where the effect of different loss functions on the logits of the network were analysed [9] and extends to the other two related losses (mixed and gdsc).

If we zoom out and analyse the other losses we explored in this study, we can observe that they can be easily derived from the cross-entropy loss. This loss in particular is ubiquitous in classification studies (including segmentation defined as pixel classification) due to its simplicity and its relationship to logistic regression. As opposed to the Dice loss, this function was derived to interact with sigmoid (and softmax in problems with multiple labels), leading to a linear gradient. In that sense, if we observe the curves for these losses (xent, focal1, focal2 and new), we can observe a smoother transition and increasing tendency when it comes to sensitivity over number of epochs and a higher mean precision. This suggests that false confident predictions are improved over time, as the weight updates are mostly linear with respect to the error on the prediction. Furthermore, the weights on the target class (lesions) are the most important trade-off between sensitivity and precision as observed in the differences between the focal1 and focal2 losses. Finally, even though cross-entropy is the most basic function, it is also highly reliable and can obtain similar results, even though it might have a higher number of epochs.

Overall, the positive detection of lesions voxels remains high after 25 epochs as evidenced by the qualitative examples in Figure 6 and Figure 7. However, small differences in false positives and negatives are observed depending on the loss, with a common trade-off between them.

### 4.2. The Effect of Randomness during Training

To understand the effect of randomness during training, we use five random sets that affect the initial value of the layer’s weights and the shuffling of the patches used for training at each epoch. To illustrate this effect, we plot the results of each epoch with a set of bands with the minimum, maximum and mean values over all seeds for each method in Section 3. One of the first conclusions that can be observed is the large overlap between all methods and metrics for image-wise results. As expected, this effect is made stronger on the testing images that were not seen during the training process. In that sense, while the mean curves can be easily distinguished between methods, it is also obvious that changing the random seed could lead to different conclusions if the lowest bound for the “baseline” methods is selected and the highest bound “seed” is used for a method we would like to highlight over the others. Furthermore, datasets with a low percentage of foreground voxels are more sensitive to that issue as exemplified by the curves on the LIT dataset shown in Figure 4.

### 4.3. The Discrepancy between Patch-Based Results and Image-Based Results

The most common way to analyse how the training process evolves is to monitor the training and validation losses. In the scenario where a model is trained on patches (due to memory constraints), these two curves represent the ability of the model to segment patches. Even though such measures are generally considered a good proxy for the real end goal (segmenting lesions on the whole brain), our experiments suggest that these values are not only over-optimistic but also highly misleading (especially when comparing methods). If we look at the patch-based results for both datasets, all methods have high DSC values over 0.7 (considered a good value for lesions) [24], with some losses reaching a value close to 1 (a perfect score). While this is not surprising because the model is trained with these same patches, the same metric drops drastically for the LIT dataset (the one with the lowest percentage of lesion voxels) and less so for the WMH dataset. Furthermore, our proposed loss function that obtains the lowest DSC on the training patches by a large margin obtains comparable results when evaluated on the image segmentation results reconstructed from patches as shown in Figure 3.

On a related note, while it might seem counterintuitive that the sensitivity is also affected when comparing results between patches and images, this phenomenon might be caused by the reconstruction of the final results through averaging. In particular, boundary lesion voxels might have a decreased score when considering different predictions for the same voxel in different relative positions within the patch. In fact, previous studies on CNNs have proven that while convolutions are inherently shift-invariant, pooling and padding can lead to encoding positional information during training [25,26]. In that sense, both reconstructing image segmentations from patches or performing inference directly on the whole image would lead to different results when comparing metrics on images and patches. Moreover, if overlapping patches are used during training, the loss metric might be unrealistic image-wise, as some voxels would be counted as independent occurrences more than once.

To conclude, these results suggest a few possible explanations that have also been corroborated by previous studies focusing on the DSC metric and its role as a measure of overlap that provides a balanced overview in terms of true and false predictions. On one hand, it is well known that the DSC metric is sensitive to the size of the object being analysed [23,27]. In our results, this is reflected by the difference between patch-wise and image-wise results but also by the fact that LIT results are a decimal point lower than those obtained with the WMH dataset. On the other hand, patch-wise results focus exclusively on patches with both foreground and background voxels (as background-only patches would not contribute to improve lesion segmentation and could be detrimental). In that sense, our analysis further emphasises the importance of not relying on loss values alone when evaluating model performance and to validate with metrics that are as close as possible to the desired end goal.

## 5. Conclusions

In this paper, we presented an evaluation of a set of common losses used for binary lesion segmentation on heavily imbalanced scenarios. We observed that training with sampled patches from images gives an unrealistic measure of the final model when applied to whole images. In other words, while some losses might provide a better performance on the training patches, that improvement is not reflected on the training images as a whole, due to the bias introduced during sampling. Furthermore, we have observed that random initialisation and shuffling can cause a large variance on the performance metrics, giving a range of different conclusions depending on which random seed and stopping epoch we take to evaluate the losses. However, most of the analysed losses have similarly optimal performance results (if the best stopping epoch and seed are known), even though they show clearly different trends throughout the training process depending on what the loss prioritises (e.g., an increase in positive predictions for our proposal). To conclude, heavily weighting the gradient of the foreground class gives the best results in terms of true positives, while still being able to obtain a competitive DSC metric on average, with an acceptable number of false positives.

## Figures and Tables

**Figure 1 sensors-24-01981-f001:**
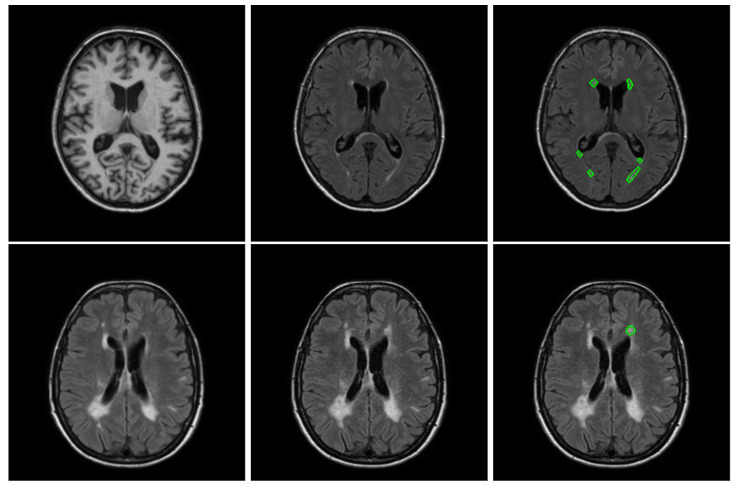
Example of slices of a randomly selected subject for each dataset. The first row depicts an example of a T1 slice, a FLAIR slice and the boundary of all the lesions for a WMH2017 subject, while the second row depicts an example of a baseline, follow-up image and the boundary of all the new lesions (not appearing on the baseline image) for a LIT longitudinal subject.

**Figure 2 sensors-24-01981-f002:**
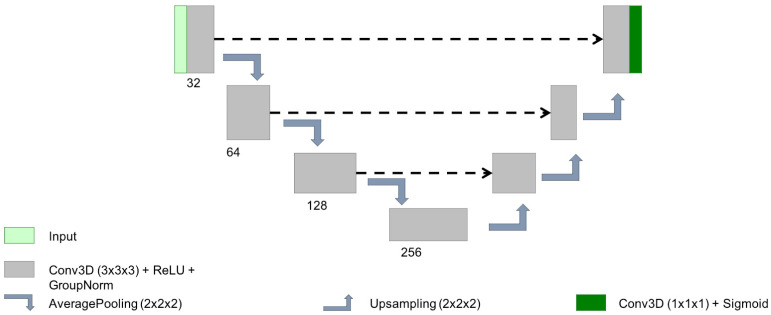
Scheme of the simple Unet architecture used for this analysis.

**Figure 3 sensors-24-01981-f003:**
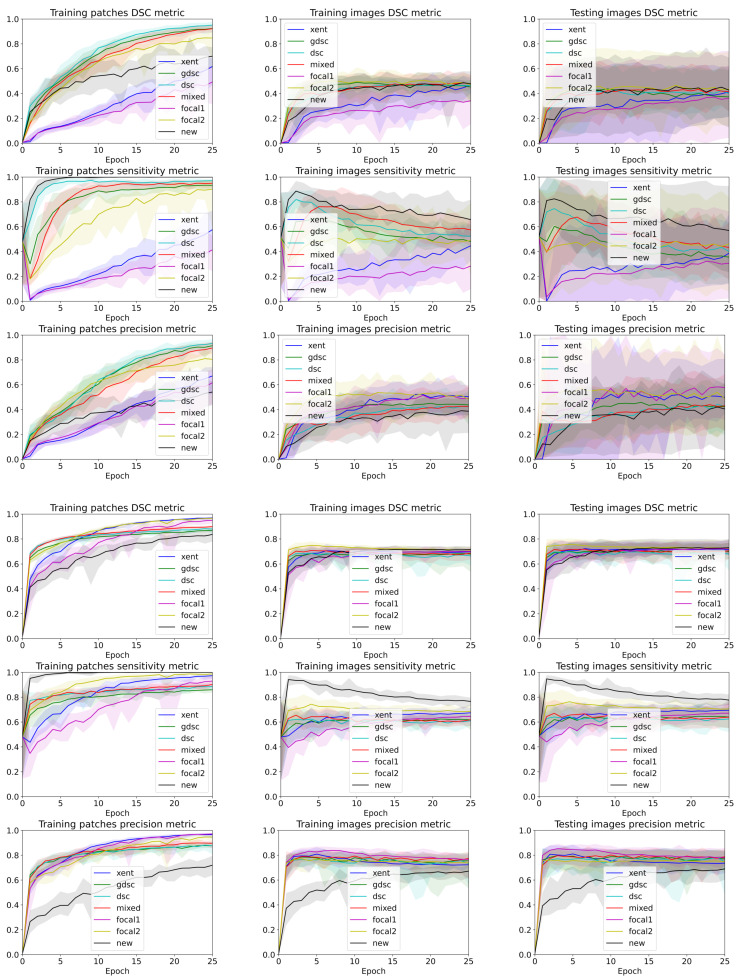
Band plots of the DSC, sensitivity and precision metric for the weights of each epoch for all the folds. The upper and lower bands represent the minimum and maximum values, while the middle line represents the mean for all random seeds and folds. The first three rows summarise the metrics for the LIT dataset, while the last three summarise the results for the WMH challenge dataset.

**Figure 4 sensors-24-01981-f004:**
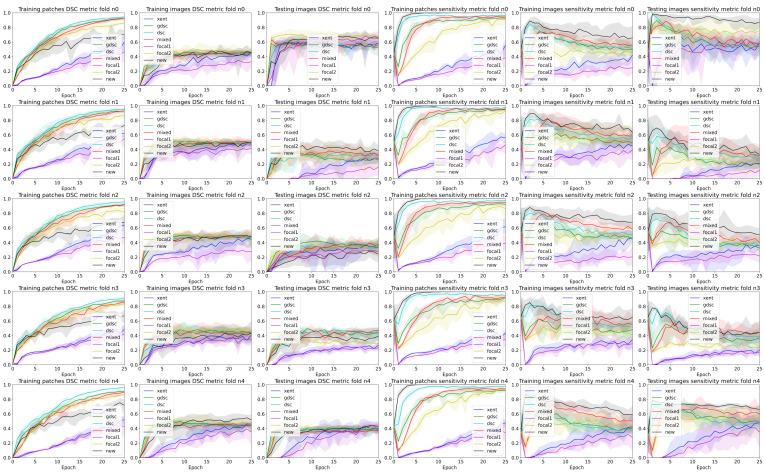
Band plots of the DSC and sensitivity metric for the weights of each epoch for each fold of the LIT dataset. The upper and lower bands represent the minimum and maximum values, while the middle line represents the mean for all the random seeds.

**Figure 5 sensors-24-01981-f005:**
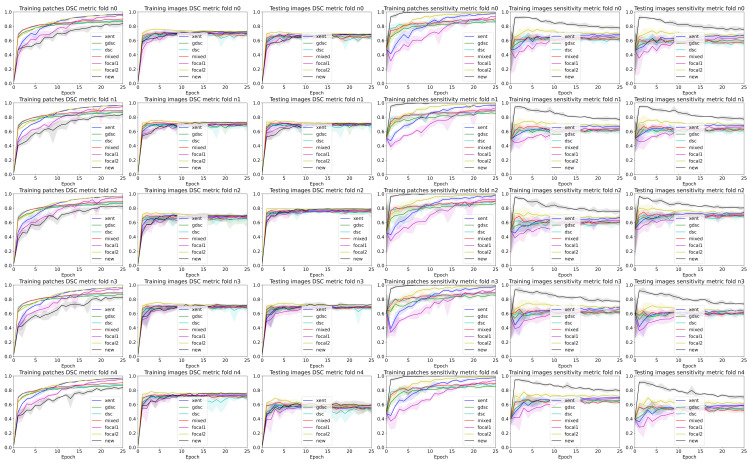
Band plots of the DSC and sensitivity metric for the weights of each epoch for each fold of the WMH dataset. The upper and lower bands represent the minimum and maximum values, while the middle line represents the mean for all the random seeds.

**Figure 6 sensors-24-01981-f006:**
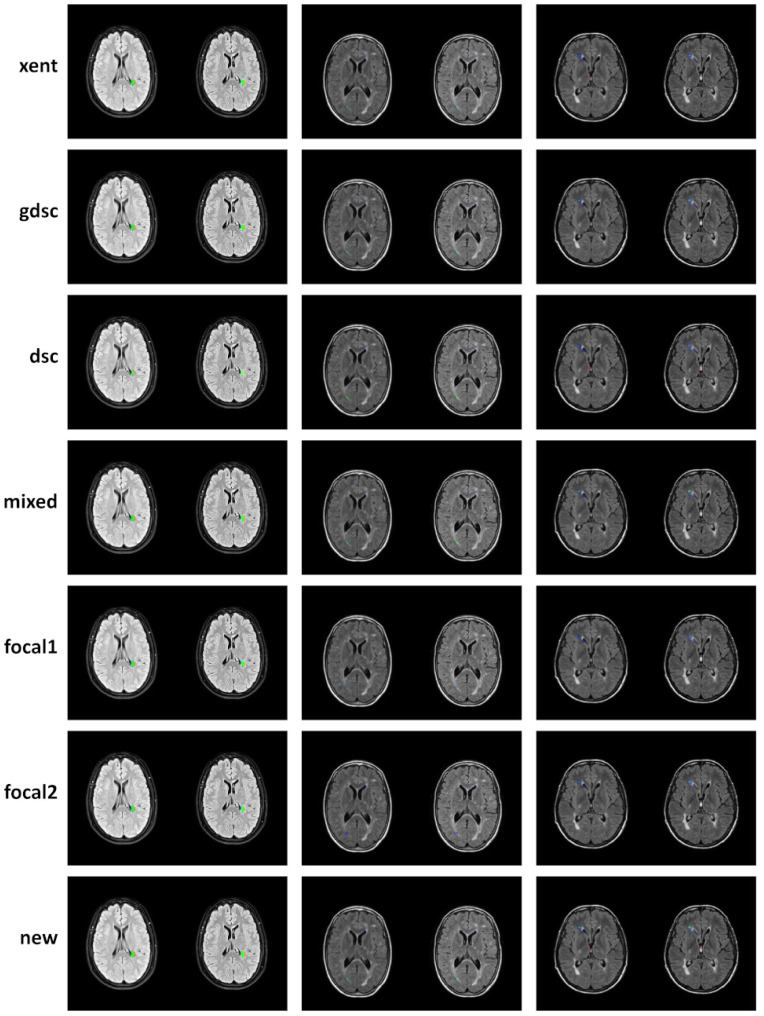
Qualitative examples of three subject from the LIT dataset where new lesions are segmented (those that appear only on the follow-up image on the right). Classified voxel are colour coded to represent true positives (green), false positives (red) and false negatives (blue).

**Figure 7 sensors-24-01981-f007:**
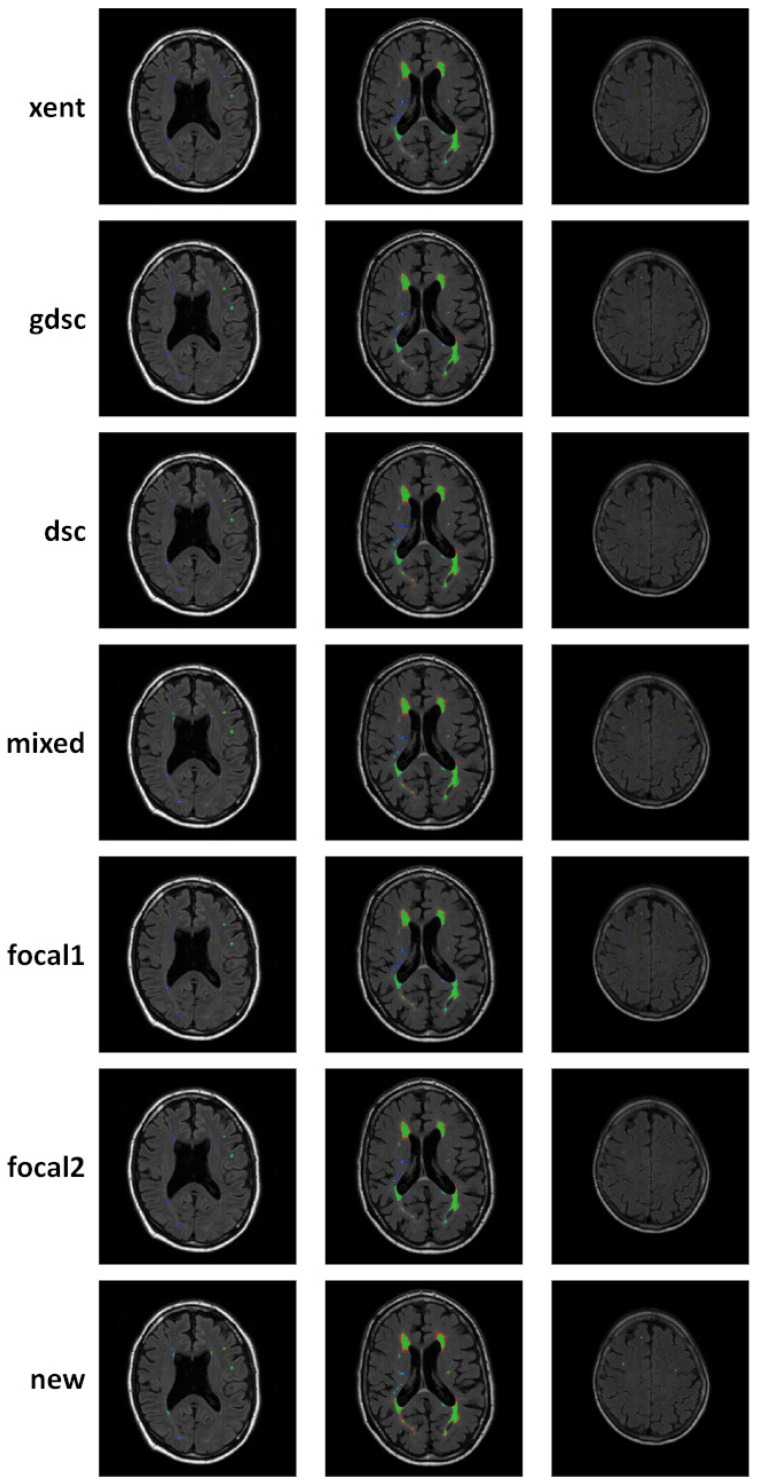
Qualitative examples of three subject from the WMH dataset, where white matter hyperintensities are segmented. Classified voxel are colour coded to represent true positives (green), false positives (red) and false negatives (blue).

**Figure 8 sensors-24-01981-f008:**
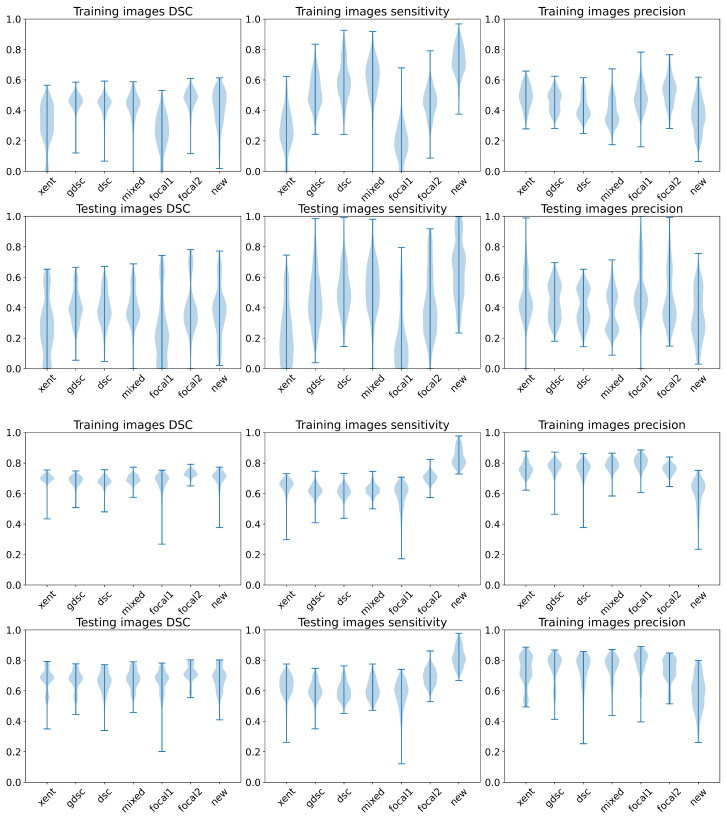
Violin plots for the DSC, sensitivity and precision metrics for all the folds and training seeds. The first row represents the results on the LIT dataset, while the second row represents the results on the WMH dataset.

**Table 1 sensors-24-01981-t001:** Statistics on the number of voxels for the foreground and background class for the analysed datasets. % (brain) refers to the lesion voxel percentage mean ± standard deviation for the whole brain, while % (batch) refers to the lesion voxel percentage for all the batches. Numbers in brackets represent the global percentage for the whole dataset and batches.

Dataset	% (Brain)	% (Batch)	Mean Lesion Size	Lesion Number
WMH2017	1.54 ± 1.48 [1.57]	1.53 ± 2.40 [1.73]	111.1750	3679
LIT longitudinal	0.14 ± 0.18 [0.15]	0.48 ± 0.84 [0.51]	116.13	156

**Table 2 sensors-24-01981-t002:** Summary of the measures for the average of the best results for each fold. Patch and train refer to the patch-wise and image-wise measures for the training set, while test stands for the image-wise measures for the testing set.

Loss	DSC			Sensitivity (${\mathrm{TP}}_{f}$)			Precision (*P*)		
	**Patch**	**Train**	**Test**	**Patch**	**Train**	**Test**	**Patch**	**Train**	**Test**
LIT longitudinal dataset									
xent	0.64	0.51	0.46	0.62	0.55	0.49	0.54	0.48	0.46
gdsc	0.93	0.54	0.53	0.96	0.81	0.77	**0.82**	0.20	0.29
dsc	**0.96**	0.55	0.52	0.99	0.89	0.84	0.54	0.29	0.24
mixed	0.93	0.55	0.54	0.98	0.86	0.82	0.74	0.18	0.24
focal1	0.59	0.48	0.46	0.52	0.49	0.49	0.55	**0.53**	**0.49**
focal2	0.89	**0.60**	**0.56**	0.97	0.70	0.63	0.78	0.48	0.38
new	0.79	0.58	**0.56**	**1.00**	**0.94**	**0.90**	0.36	0.14	0.23
WMH challenge 2017									
xent	**0.96**	0.68	0.42	0.95	0.60	0.55	**0.97**	**0.80**	0.47
gdsc	0.86	0.71	0.69	0.85	0.64	0.63	0.88	0.79	0.76
dsc	0.88	0.56	0.54	0.87	0.55	0.65	0.89	0.60	0.60
mixed	0.89	0.56	0.44	0.89	0.58	0.63	0.90	0.62	0.47
focal1	0.94	0.71	0.68	0.91	0.64	0.59	**0.97**	**0.80**	**0.81**
focal2	**0.96**	**0.72**	**0.72**	0.98	0.72	0.69	0.94	0.73	0.75
new	0.79	0.70	0.71	**1.00**	**0.85**	**0.81**	0.65	0.61	0.63

## Data Availability

The data for this study are publicly available on the websites of the challenge organisers. The code used for this study is also publicly available at https://github.com/marianocabezas/rethinking_dsc (accessed on 1 January of 2024).

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
