# Peer review of "An Analysis of Loss Functions for Heavily Imbalanced Lesion Segmentation"

_sensors, 2024, doi:10.3390/s24061981_

Round 1

Reviewer 1 Report

Comments and Suggestions for Authors

This article analyzes different common loss indicators for image segmentation in severely imbalanced scenarios. The 3D Unet architecture was chosen in the manuscript and evaluated on two publicly available datasets. The idea is interesting, but there are still the following problems to be solved:

1. Please cite some references in the second paragraph of Section 1 to verify the source and credibility of the information.

2. Please move Table 1 to Section 2 and enlarge Table 1. Annotations for tables should be written in the main text rather than in the table title.

3. Please correct the symbol "??" in the last paragraph of Section 1。

4. In Section 2, please select photos from different datasets for reference.

5. Please explain these hyperparameters in section 2.2 and demonstrate why they were chosen in this way

6. Enlarge Table 2 to improve the readability.

7. there are a large number of curves in Figures 3 and 4, but the explanation of these images is not detailed enough.

Reviewer 2 Report

Comments and Suggestions for Authors

Please see the file attached.

Round 2

Reviewer 2 Report

Comments and Suggestions for Authors

Please see the file attached.

Author Response

Please see the file attached.

Round 3

Reviewer 2 Report

Comments and Suggestions for Authors

I thank the authors for their revisions to their manuscript. However, there are still some issues to be addressed.

- Thank you for adding Figs. 6 and 7. However, these figures are not referenced and discussed in the text. It often happens that the quantitative evaluation and the visualization results do not match sensibly (i.e., the quantitative results are good, but the visualization do not reach the desired level). Thus, I think it is important to discuss what features of lesions each model tends to be good or bad at capturing based on the visualization results. Please add the description of results and the discussion.

In Fig. 8, the precision result of testing images in the WMH dataset is the same as the sensitivity result. Please correct it.

Author Response

We thank the reviewer again for the thorough review and feedback, and we have updated our manuscript accordingly (only changes for the current revision are highlighted in blue in the new manuscript). Finally, here are our answers to the comments:

- Thank you for adding Figs. 6 and 7. However, these figures are not referenced and discussed in the text. It often happens that the quantitative evaluation and the visualization results do not match sensibly (i.e., the quantitative results are good, but the visualization do not reach the desired level). Thus, I think it is important to discuss what features of lesions each model tends to be good or bad at capturing based on the visualization results. Please add the description of results and the discussion.

We thank the reviewer for the suggestion. We agree that it is helpful to not only have reference to the figures on the text, but also explanations about them. To that end, we have included a few sentences on the results and discussion sections to highlight how the qualitative examples align to the qualitative results.

In Fig. 8, the precision result of testing images in the WMH dataset is the same as the sensitivity result. Please correct it.

Thank you for pointing that out. We have now corrected the figure to show the right violin plot.